# Prevalence of tick infestation and molecular characterization of spotted fever *Rickettsia massiliae* in *Rhipicephalus* species parasitizing domestic small ruminants in north-central Nigeria

**Nusirat Elelu[1], Shola David Ola-Fadunsin[2]\*, Adefolake Ayinke Bankole[3], Mashood Abiola Raji[4], Ndudim Isaac Ogo[5], Sally Jane Cutler[3]**

1 Department of Veterinary Public Health and Preventive Medicine, Faculty of Veterinary Medicine, University of Ilorin, Ilorin, Nigeria, 2 Department of Veterinary Parasitology and Entomology, Faculty of Veterinary Medicine, University of Ilorin, Ilorin, Nigeria, 3 School of Health, Sport and Bioscience, University of East London, London, United Kingdom, 4 Department of Veterinary Microbiology, Faculty of Veterinary Medicine, University of Ilorin, Ilorin, Nigeria, 5 National Veterinary Research Institute, Vom, Nigeria

\* olafadunsin.sd@unilorin.edu.ng

**Data Availability Statement:** All relevant data are within the paper and its Supporting information files.

## Abstract

Ticks are of great menace to animal and human health. They serve as vectors to both animals and human pathogens including *Rickettsia* species. Tick-borne rickettsiosis in West Africa remains incompletely understood. We determined the prevalence of tick infestation among small ruminants and molecularly described a clinically significant spotted fever *Rickettsia massiliae* from *Rhipicephalus* ticks collected from North-Central, Nigeria. A total of 352 small ruminants comprising of 152 sheep and 200 goats that were brought for slaughter at the major small ruminant slaughterhouse in Ilorin were examined for the presence of ticks. The collected *Rhipicephalus* species were subjected to molecular studies to detect and characterize *Rickettsia massiliae*. Of the small ruminants examined, 21 sheep and 46 goats were infested with ticks representing 13.82% and 23.00% respectively. Eight and nine different species of ticks were detected in sheep and goats respectively, with *Rhipicephalus* (*Boophilus*) *decoloratus* being the most prevalent tick species in both sheep and goats. There was a significant difference (p <0.01) in the prevalence of the different tick species collected in sheep and in goats. Based on the PCR amplification of the 23S-5S intergenic spacer (IGS), only 2 of the 142 *Rhipicephalus* tick samples screened for *R. massiliae* were positive (1.41%; 95% CI = 0.39–4.99). *Rickettsia massiliae* was detected from *Rhipicephalus turanicus* collected from sheep. Sequences obtained from the PCR carried out by amplifying *Rickettsia* 23S-5S IGS showed 99–100% close identity with members of the *R. massiliae* group. This study has for the first time confirmed the presence of spotted fever group *Rickettsia massiliae* from feeding ticks in Nigerian small ruminants. Further investigations to determine the possible pathogenic role of human *R. massiliae* infection in Nigeria would be beneficial.

**Funding:** The funders had no role in study design, data collection and analysis, decision to publish, or preparation of the manuscript. This study was supported by the Africa Research Excellence Fund Fellowship. Grant Number: MRF-157-0022-F-ELELU. The grant recipient is Dr. Nusirat Elelu.

**Competing interests:** The authors have declared that no competing interests exist.

## Introduction

Small ruminants (sheep and goat) rearing is one of the most important aspects of agriculture in most parts of the world [1]. This has contributed greatly to the growth and development of many economies worldwide [1, 2]. Small ruminants are a major component of the ruminant industry in Nigeria, with an estimated population of sheep and goats at 22.1 million and 34.5 million, respectively [3, 4]. Small ruminants maintain an available economic and ecological niche in Nigerian agriculture, as they represent about one-third of the country's agricultural gross domestic product [3, 5]. Sheep and goats are very vital sources of protein to man in terms of meat and milk in both developed and developing economies [4, 6], they are also useful for the provision of manure, skin, and for household income and socio-cultural purposes [4, 7].

Ticks are the second most important arthropod parasite after mosquitoes, that affect mammals and birds [8, 9]. They are ranked as the most economically important ectoparasites of livestock in the tropics, including sub-Saharan Africa [10, 11]. A number of tick species can act as vectors of pathogens causing a number of tick-borne diseases, which causes a serious impairment to the health, welfare, production, and reproduction of ruminants including sheep and goats [9, 12].

Tick-borne rickettsiosis is among the oldest known vector-borne zoonotic diseases in the world [13]. Although still largely neglected, human rickettsiosis is the second most frequent cause of febrile illnesses after malaria in travelers returning from sub-Saharan Africa [14]. Several human tick-borne *Rickettsia* are classified as members of the spotted fever groups (SFG). So far, the recognized species of the SFG include: *R. conorii*, the agent of Mediterranean spotted fever; *R. rickettsii*, agent of Rocky Mountain spotted fever; *R. africae*, agent of African tick bite fever; *R. sibirica*, agent of Siberian tick typhus. Others include: *R. slovaca*, *R. honei*, *R. japonica*, *R. australis*, *R. akari*, *R. felis*, *R. helvetica*, *R. parkeri*, *R. peacockii* as well as the newly emerging human pathogen: *R. massiliae* and its closely related species that make up the *R. massiliae* group which consists of *R. massiliae*, *Rickettsia* sp. Bar29, *R. rhipicephali*, *R. motanensis* and *R. aeschlimannii* [13, 15–17].

*Rickettsia massiliae*, although first isolated from ticks in Marseille, is a confirmed human pathogen [16]. Subsequently, human cases of spotted fever group *R. massiliae* infection have been reported in Europe [18]; and South America [19]. Members of the *R. massiliae* group have been established to be some of the *Rickettsia* species that are naturally resistant to rifampicin by a Phe-to-Leu mutation within the rpoB gene [20]. In Europe and Africa, members of the *Rhipicephalus* tick complex are the documented vectors of *R. massiliae* [21]. This tick transmitting zoonotic pathogens has been reported to cause fever, palpable purpuric rash on the upper and lower extremities, skin lesion (eschar) on the right leg in humans [19]. Chorioretinitis with macular involvement was also reported in human cases with *R. massiliae* [22].

Limited molecular studies exist describing ticks and associated *Rickettsia* spp. affecting cattle and dogs in Nigeria [23, 24]. In fact, only a single study reported *R. massiliae* in questing *Rhipicephalus evertsi* ticks collected from vegetations in Southern Nigeria [25]. In contrast, no such studies have been conducted on feeding ticks among small ruminants from northern Nigeria. We report our findings in small ruminants from North-Central Nigeria using sensitive molecular methods to identify the tick-borne *Rickettsia* species present in small ruminants.

## Materials and methods

### Study location

Ethical approval for the study was granted by the University of Ilorin ethical review committee (Approval number: UERC/ASN/2018/1387).

This study was conducted at the Ipata municipal abattoir in Ilorin, Kwara State, North-Central Nigeria. This abattoir is one of the largest small ruminant abattoirs in the sub-region.

## Collection and identification of ticks

Between May and August 2019, 352 apparently healthy small ruminants (152 sheep and 200 goats) were screened for the presence of ticks. Two hundred and forty adult ticks were collected from 67 small ruminants (21 sheep and 46 goats). Ticks were collected at the point of slaughter with the use of forceps thorough body search. The collected ticks were placed in 70% ethanol and transported to the laboratory. All of the ticks were morphologically identified by previous reported standard keys (based on: size, mouthparts, scutum, presence or absence of pale rings on legs, presence or absence eyes, etc.) as documented by Mathison and Pritt [26] and Walker et al. [27].

## Deoxyribonucleic acid (DNA) extraction and PCR amplification

Genomic DNA was extracted from each tick that was morphologically identified as *Rhipicephalus* species (been the predominant tick species; n = 142) using the QIAamp tissue DNA kit (Qiagen, Hilden, Germany), according to the protocols prescribed by the manufacturers. Before the extraction of DNA, the individual tick was washed with phosphate-buffered saline (PBS), then air-dried for about 10 min on tissue paper and the tick was separately sliced into small pieces by using a sterile scalpel blade, afterward, it was manually homogenized with a sterile micro pestle, and resuspended in 200 μl of lysis buffer and 20 μl of proteinase K and incubated overnight at 56°C with continuous gentle shaking. The eluted ticks DNA was screened for the presence of *Rickettsia* by polymerase chain reaction (PCR) targeting the 350 bp DNA non-coding region of the *Rickettsia* 23S-5S intergenic spacer (IGS), and the PCR preparations were done as described by Kakumanu et al. [28] with modifications whereby only the secondary primer sets were used in a final volume of 25 μl. DNA from *R. hoogstraali* served as positive control while nuclease-free water was used as negative control in all PCR reactions. The set of oligonucleotide primers, the targeted fragment, thermocyclic protocols, and amplicon sizes (bp) for this study are summarized in Table 1.

The resultant amplicons were electrophoresed and visualized on 1.5% agarose gel that was stained with ethidium bromide ethidium to check the quality of amplification.

**Table 1.** *Rickettsia massiliae* oligonucleotide primers used for PCR amplification and its thermocyclic protocols.

| Primer Name | Primer sequence (5′–3′) | Fragment | Thermocyclic protocols | Amplicon size (bp) |
|---|---|---|---|---|
| RCK/23-5N1F | TGTGGAAGCACAGTAATGTGTG | 23S-5S IGS | ID: 95°C / 10 min | 350 |
| | | | (No of repeat: 1) | |
| RCK/23-5N1R | TCGTGTGTTTCACTCATGCT | | D: 94°C / 30 sec | |
| | | | A: 56°C / 30 sec | |
| | | | E: 72°C / 90 sec | |
| | | | (No of cycles 35) | |
| | | | FE: 72°C / 10 min | |
| | | | HT°: 4°C | |

Themocyclic protocol is as used for this study.

ID = initial denaturing of DNA, D = denaturing of DNA, A = annealing of primers, E = extension of DNA, FE = finial extension of DNA, HT° = holding temperature, bp = base pair.

## Gel purification and confirmation of the 23S-5S intergenic spacer (IGS) region of *Rickettsia* groups

Representative positive amplicons from the PCR were purified for sequencing using the Qiaquick® PCR purification kit (Qiagen®) according to the manufacturer's instructions. Sequencing of the PCR products and confirmation of the 23S-5S intergenic spacer (IGS) region of *Rickettsia* groups was carried out at DBS Genomics, Durham.

## Data and phylogenetic analyses

The Statistical Package for Social Sciences (SPSS) version 23.0 (SPSS Inc., Chicago, Illinois) was used for the statistical analysis. The prevalence and corresponding 95% confidence interval (CI) were used to measure the level of infestation among sheep and goats. The Chi-square ($\chi^2$) test was used to evaluate the level of each tick species infestation on sheep and goats. Statistical significance was set at $p < 0.05$. Sequences obtained from this study were searched for homologous sequences in the GenBank® using BLASTn (www.ncbi.nlm.nih.gov/BLAST). Sequences were aligned by using the Clustal W program. Based on these alignments, nucleotide alignments were performed and phylogenetic analyses were conducted in MEGA version 7.0 (https://www.megasoftware.net/). The phylogenetic tree was constructed by the Maximum Likelihood method, 1000 replicates bootstrap. The sequence obtained from this study has been deposited in GenBank® under the accession number OK350078.

## Results

Of the 152 sheep and 200 goats sampled, 21 sheep and 46 goats were infested with ticks, representing 13.82% and 23.00% respectively. Sixty-seven of the total 352 small ruminants sampled were infested with ticks. The prevalence of tick infestation was significantly higher in goats compared to sheep ($\chi^2$ = 4.73; p = 0.03) (Table 2).

Eight different tick species (belonging to 3 genera) were detected in sheep, while 9 species (belonging to 4 genera) were detected in goats. Eighty-one ticks were collected from sheep, while 159 ticks were collected from goats.

*Rhipicephalus* species were the most numerous species infesting sheep (55/81; 67.90%; 95% CI = 57.17–77.37) and goats (87/159; 54.72%; 95% CI = 46.93–62.33). In sheep, *Rhipicephalus* (*Boophilus*) *decoloratus* (n = 22; 27.16%) was the most prevalent tick species, followed by *Amblyomma variegatum* (n = 15; 18.52%), while *Rhipicephalus lunulatus* was the least prevalent with a 2.47%. *Rhipicephalus* (*Boophilus*) *decoloratus* and *Amblyomma variegatum* were the most prevalent tick species in goats, while *Rhipicephalus evertsi* was the least prevalent. There was a significant difference (p <0.01) in the prevalence of the different tick species collected in sheep and in goats (Table 3).

**Table 2. Prevalence of tick infestation among small ruminants in Ilorin, Nigeria.**

| Small ruminants | Number sampled | Number infested | Prevalence (%) | 95% CI |
|---|---|---|---|---|
| Sheep | 152 | 21 | 13.82 | 9.00–20.00 |
| Goats | 200 | 46 | 23.00 | 17.56–29.21 |
| **Total** | **352** | **67** | **19.03** | **15.19–23.39** |

CI = Confidence interval.

$\chi^2$ (Chi Square value) = 4.73.

DF (Degrees of Freedom) = 1.

p-value = 0.03.

**Table 3. Diversity and prevalence of tick species infesting small ruminants in Ilorin, Nigeria.**

| Tick species | Number (%) | $\chi^2$ | DF | p-value |
|---|---|---|---|---|
| **Sheep (n = 81)** | | | | |
| *Rhipicephalus turanicus* | 5 (6.17) | 32.61 | 7 | <0.01[¥] |
| *Rhipicephalus sanguineus* | 12 (14.81) | | | |
| *Rhipicephalus lunulatus* | 2 (2.47) | | | |
| *Rhipicephalus evertsi* | 5 (6.17) | | | |
| *Rhipicephalus (Boophilus) decoloratus* | 22 (27.16) | | | |
| *Rhipicephalus (Boophilus) geigyi* | 9 (11.11) | | | |
| *Amblyomma variegatum* | 15 (18.52) | | | |
| *Hyalomma rufipes* | 11 (13.58) | | | |
| **Goats (n = 159)** | | | | |
| *Rhipicephalus turanicus* | 2 (1.26) | 167.10 | 8 | <0.01[¥] |
| *Rhipicephalus sanguineus* | 27 (16.98) | | | |
| *Rhipicephalus lunulatus* | 3 (1.89) | | | |
| *Rhipicephalus evertsi* | 1 (0.63) | | | |
| *Rhipicephalus (Boophilus) decoloratus* | 51 (32.08) | | | |
| *Rhipicephalus (Boophilus) geigyi* | 3 (1.89) | | | |
| *Amblyomma variegatum* | 39 (24.53) | | | |
| *Hyalomma truncatum* | 15 (9.43) | | | |
| *Hyalomma rufipes* | 18 (11.32) | | | |

n = Number of ticks collected in each small ruminant.

$X^2$ = Chi square.

DF = Degrees of Freedom.

[¥] = Significant at p < 0.05.

Based on the PCR amplification of the 23S-5S IGS, 2 of the 5 *Rhipicephalus turanicus* tick species collected from sheep were positive for *R. massiliae* (40.00%; 95% CI = 11.76–76.93). The molecular prevalence of *R. massiliae* in relation to the total number of *Rhipicephalus* tick (142) screened was 1.41%; 95% CI = 0.39–4.99. Sequences obtained from the PCR carried out by amplifying *Rickettsia* 23S-5S intergenic spacer (IGS) showed 99% close identity with members *R. massiliae* group. Phylogenetic analysis carried out based on the IGS showed that samples from this study clustered together with other reported *R. massiliae* available in the gene bank (Fig 1).

## Discussion

The overall ticks prevalence of 19.03% we recorded among small ruminants in this study is higher than the 8.10% total prevalence reported among small ruminants in Makurdi, North-Central Nigeria [29] and the 14.85% reported in Uli, southeast Nigeria [5]. Although a higher total prevalence of tick infestation among small ruminants has been documented in Pakistan (51.02%) [30] and Ethiopia (79.70%) [31]. The differences could be attributed to seasonal, environmental, and ecological factors.

A higher prevalence of tick infestation was observed in goats compared to sheep. This may be linked to the more roaming nature of goats compared to sheep, making goats more exposed to questing ticks in vegetations. It could also be due to the high wool level of sheep, making it difficult for ticks to attach on them.

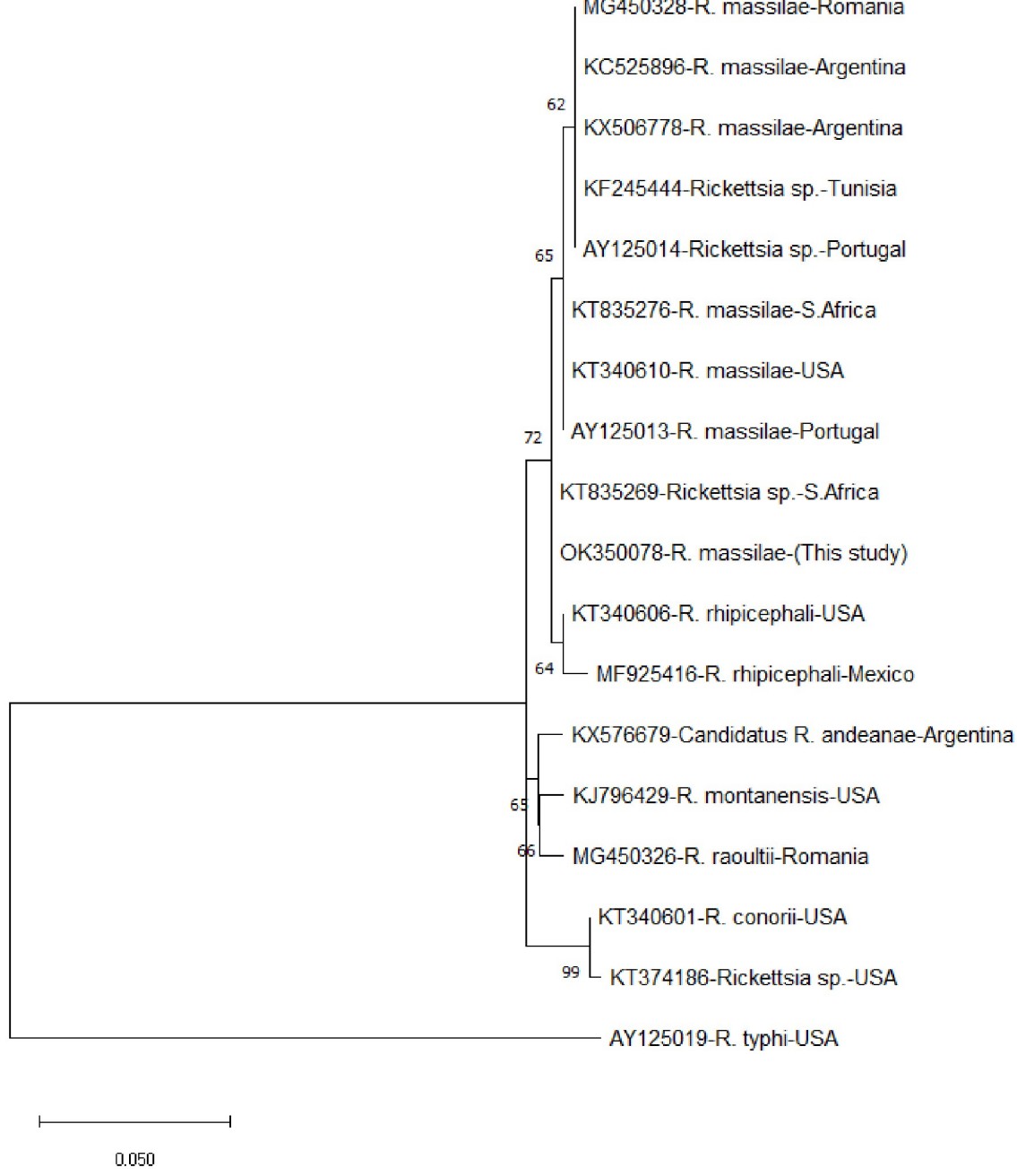

**Fig 1. Molecular phylogenetic analysis inferred by Maximum Likelihood of *Rickettsia massiliae* based on the 350bp partial 23S-5S IGS sequences of *Rickettsia* species taken from the NCBI database and sequence amplified from *Rhipicephalus* species tick collected from Nigerian small ruminants.**

Ticks are usually present in vegetations, awaiting suitable host(s) to attach onto [32], thus goats become more vulnerable due to their more roaming nature. Higher prevalence of tick infestation has been reported in goats compared to sheep in Nigeria [33] and outside Nigeria (Pakistan) [30].

Eight different species of ticks were detected among sheep in our study. This number is higher than the two, five, and six different species of ticks detected among sheep in Nigeria by

Alayande et al. [34], Kaze et al. [33] and Maidala [35] respectively. Lower number of tick species (4) have been reported to infest sheep in Ethiopia [36]. In like manner, a lower number of tick species was reported among goats in Nigeria [33, 34, 35] and outside Nigeria (Ethiopia) [31] compared to the number (9) we observed in our study. These observations may suggest that tick infestation is of concern among small ruminants in the study area.

*Rhipicephalus* was the most predominant genus of ticks infesting sheep and goats, with *Rhipicephalus* (*Boophilus*) *decoloratus* being the most prevalent species among both animal species in our study. *Rhipicephalus* species has been reported to be the most prevalent genus infesting small ruminants in Nigeria [1, 29, 33, 35] and Ethiopia [37]. The high prevalence of *Rhipicephalus* species among other tick genera was the reason we used the genus in our molecular study of *Rickettsia massilae*.

The 23S-5S IGS is a conserved, non-coding region of DNA that is useful in *Rickettsia* molecular taxonomy [28]. Here, we are able to confirm the presence of *Rickettsia massilae* in ticks collected from domestic small ruminants. These tick-borne pathogens appear common in South Africa and have been reported from Cameroon [38], Central African Republic [39], and Ivory Coast [40].

The 40.00% molecular prevalence of *R. massiliae* among *Rhipicephalus turanicus* ticks collected from sheep in our study is far higher than the 3.51% of *Rickettsia massiliae* reported among *Rhipicephalus turanicus* ticks collected from sheep in China [41]. This significant level of high molecular prevalence of *R. massiliae* among *Rhipicephalus turanicus* collected from sheep in our study, calls for great public health concerns. *Rickettsia massiliae* is a known pathogenic *Rickettsia* causing spotted fever in humans [42].

*Rickettsia massiliae* was detected only from *Rhipicephalus turanicus* collected from sheep in the study area. In a similar vein, Wei et al. [41] and Ereqat et al. [43] detected *R. massiliae* in *Rhipicephalus turanicus* infesting sheep in China and the Palestinian territories respectively. *Rhipicephalus turanicus* is a three-host tick [27], with both transovarial and transstadial transmission of *R. massiliae* reported in the tick vector [44]. This shows there may be possibilities of *R. massiliae* infection in other animals infested by this tick species during the developmental stages of its lifecycle. There was a close relationship seen with the *R. massiliae* of this study and that seen in South Africa, suggesting a universal spread of the organism.

*Rickettsia massiliae*, strain Bar29, has been previously reported to be detected in engorged female *Rhipicephalus turanicus* tick collected in Corsica [44] and from many other members of the *Rhipicephalus* ticks from different parts of the world [21]. *Rickettsia massiliae* was previously isolated from questing *R. evertsi* ticks in cattle grazing areas from Southern Nigeria [25]. However, *Ricketssia massiliae* has not been reported in ticks infesting highly domesticated sheep and goats kept in Nigeria.

Detection of *R. massiliae* in feeding tick on sheep is of zoonotic concern. This is of significance because, sheep are one of the most widespread animals kept by humans in sub-Saharan Africa owing to their high fertility, short generation interval, adaptation to harsh environments, and are considered as a source of investments for rural households [45]. They are mostly kept as free-range and often tethered at night close to human dwellings. This provides a suitable environment for both contacts with disease vectors such as ticks thus facilitating their transfer of infections including *Rickettsia massiliae* to humans.

## Conclusions

*Rhipicephalus* species was the most prevalent tick genus infesting small ruminants in the study area, with *Rhipicephalus* (*Boophilus*) *decoloratus* being the most prevalent species in both sheep and goats. This study has for the first time confirmed the presence of the spotted fever

group *Rickettsia massiliae* from feeding ticks in Nigerian small ruminants. *Rickettsia massiliae* was detected in two *Rhipicephalus turanicus* ticks collected from sheep. So far it is not recognized as a potential human pathogen in Nigeria and is not likely to be considered during the evaluation of clinical cases. Confirmed life-threatening human cases elsewhere in the world emphasize the need to consider this diagnosis and investigate among clinical patients most at risk. Further investigations with more extensive tick samples as well as human studies determining the possible pathogenic role of human *R. massiliae* infection in Nigeria would be beneficial.

## Supporting information

**S1 Data.**
(XLSX)

## Acknowledgments

We would also like to acknowledge Mr. O. Akintola for providing technical support for the project.

## Author Contributions

**Conceptualization:** Nusirat Elelu, Shola David Ola-Fadunsin.

**Data curation:** Nusirat Elelu, Shola David Ola-Fadunsin.

**Formal analysis:** Nusirat Elelu, Shola David Ola-Fadunsin.

**Funding acquisition:** Nusirat Elelu.

**Investigation:** Nusirat Elelu, Shola David Ola-Fadunsin, Adefolake Ayinke Bankole.

**Methodology:** Nusirat Elelu, Shola David Ola-Fadunsin, Adefolake Ayinke Bankole, Ndudim Isaac Ogo.

**Supervision:** Mashood Abiola Raji, Sally Jane Cutler.

**Visualization:** Nusirat Elelu.

**Writing – original draft:** Nusirat Elelu, Shola David Ola-Fadunsin.

**Writing – review & editing:** Nusirat Elelu, Shola David Ola-Fadunsin, Ndudim Isaac Ogo, Sally Jane Cutler.

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
