## [Decision Letter · Decision Letter 0]

14 Nov 2021

PONE-D-21-31323Prevalence of tick infestation and Molecular characterization of spotted fever Rickettsia massiliae in Rhipicephalus species parasitizing domestic small ruminants in North-Central NigeriaPLOS ONE

Dear Dr. Ola-Fadunsin,

Thank you for submitting your manuscript to PLOS ONE. After careful consideration, we feel that it has merit but does not fully meet PLOS ONE’s publication criteria as it currently stands. Therefore, we invite you to submit a revised version of the manuscript that addresses all of the points raised during the review process.

We look forward to receiving your revised manuscript.

Kind regards,

Brian Stevenson, Ph.D.

Academic Editor

PLOS ONE

Journal Requirements:

2. Thank you for including the following ethics statement on the submission details page:

'University of Ilorin, Nigeria.'

Please amend your current ethics statement to confirm that your named institutional review board or ethics

committee specifically approved this study.

Reviewers' comments:

Reviewer's Responses to Questions

**Comments to the Author**

1. Is the manuscript technically sound, and do the data support the conclusions?

Reviewer #1: Partly

Reviewer #2: No

Reviewer #3: Partly

Reviewer #4: Yes

2. Has the statistical analysis been performed appropriately and rigorously? 

Reviewer #1: No

Reviewer #2: Yes

Reviewer #3: Yes

Reviewer #4: Yes

3. Have the authors made all data underlying the findings in their manuscript fully available?

Reviewer #1: Yes

Reviewer #2: Yes

Reviewer #3: No

Reviewer #4: No

4. Is the manuscript presented in an intelligible fashion and written in standard English?

Reviewer #1: Yes

Reviewer #2: Yes

Reviewer #3: Yes

Reviewer #4: Yes

5. Review Comments to the Author

Reviewer #1: It will be good to see other statistical analysis on the manuscript. The authors need to use scientific words. There must also be consistency throughout the manuscript (structure, language used). It would be good to also get result on the other ticks species that were collected, even though in low numbers but they must also be tested for the pathogen. For interest sake were there any other pathogens tested or any co infection observed.

Reviewer #2: Nusirat Elelu., et al, reported in the present manuscript entitled: “Prevalence of tick infestation and Molecular characterization of spotted fever Rickettsia massiliae in Rhipicephalus species parasitizing domestic small ruminants in North Central Nigeria” a survey on tick infesting sheep and goats in the Ipata municipal. The authors have collected the tick from the animals at the local slaughterhouse. The authors report that ticks belonging the Rhipicephalus genre were the most collected from the small ruminants. PCR amplification targeting the 350 bp DNA of the Rickettsia 23S-5S intergenic spacer followed by sequencing have revealed the presence of R. massiliae in two Rhipicephalus turanicus ticks. The data collected and presented in the manuscript lack originality. All the results have been already reported in Nigeria regarding the presence of the different tick species identified in the study. In addition, R. massiliae were also reported in Nigeria. Albeit, the authors claim that the manuscript is the first report of the amplification of the DNA of R. massiliae from tick infesting small animals. Even if the primers used in the study targeting the 23S-5S region of Rickettsia were reported able to amplify the DNA of R. massiliae, the shortness of the amplicon (350 bp DNA) is a limitless to compare between related species of Rickettsia. Using other genes like gltA and rpoB were suitable to clarify the genetic position of the species amplified in these study compared to other closet species. Finally, it was suitable that the authors investigate the presence the presence of Rickettsia species in the other tick species rather than Rhipicephalus ticks. Authors can also consider a molecular survey of other related Rickettsiaceae species like Anaplasmataceae to add more valuable data.

Minor revision

Add tick identification methodology to the material and method section.

Reviewer #3: Major comments

- Why was DNA extracted only in ticks that were identified as Rhipicephalus? Number of ticks is not a scientific reason. Hyalomma ticks can also be infected with Rm.

- The bootstrap values in the Rickettsia massieliae grouping is really poor. Only 40, which in most cases would not be reported. This may be due to using different lengths of sequences. Were the sequences trimmed to represent the same 350 bp piece amplified in the PCR? Also, how do the authors explain that their potential strain is closer to USA strains than strains in Tunisia.

- The authors should show the alignment and blast results. The authors do not report performing any BLAST analysis, however, this should be the first step before moving into the ClustalW alignment and MEGA analysis.

- Likewise, the sequences obtained after Sanger sequencing should be deposited into NCBI and the Genbank numbers should be made available to the scientific community.

- The role of sheep as “reservoir” host is a stretch (see minor comments). Unless, several animals are found infected. The significance of two infected ticks in whether an animal may serve as a “reservoir” is unknown. That statement should be deleted.

- Although the reviewer believes that studies in the epidemiology of diseases are very beneficial. It would be better to start by determining the actual incidence of the pathogen among ticks in the area since only two ticks were found.

Minor comments

- Although the issues in grammar and English are not substantial, the reviewer recommends that the authors consult a colleague for editing.

- Lane 75 – rather than regarded, the Rickettsia belong to the SFG or are classified.

- Be consistent, sometimes the authors use “μl” and sometimes “ul”.

- Ticks tend to be seasonal. May the differences reported in this study differ from other studies due to the seasons when they were carried out?

- Lane 261. Given that only two Rht were collected from sheep and the two were infected with Rm that is a 100% infection rate. By comparing to the rest of the Rhipicephalus sp in the study, the authors are underestimating the infection rates. If fact this study has much higher infection rates that the cited work. Are these sheep from a particular farm? What information is available. Were the two ticks collected from the same animal? And what other ticks were collected from this particular animal? Was it only these two ticks? If so, was the animal infected?

Reviewer #4: This manuscript determined the prevalence of tick infestation among small ruminants and molecularly described a clinically significant spotted fever Rickettsia massiliae from Rhipicephalus ticks collected from North-Central, Nigeria, which means regional study of a pathogen and its host.

Title and keywords title must use different words.

Are the sheep where the ticks were found hairless? or not?

Discuss regarding the life cycle of these ticks if it is from one or more hosts?

Concerning the 23S-5S IGS Rickettsia molecular taxonomy it will be important to provide a database on the sequences found.

as in the reference in line 19. García-García JC .......,

review all references

6. PLOS authors have the option to publish the peer review history of their article (what does this mean?). If published, this will include your full peer review and any attached files.

Reviewer #1: No

Reviewer #2: No

Reviewer #3: No

Reviewer #4: No

---

## [Author Response · Author response to Decision Letter 0]

26 Jan 2022

Response to reviewers

Academic Editor’s comments 

Response: Comments by Academic Editor has been carried out. Figure 1 has been deleted.

Reviewer #1: 

Comment 1-It will be good to see other statistical analysis on the manuscript. The authors need to use scientific words. There must also be consistency throughout the manuscript (structure, language used). It would be good to also get result on the other ticks species that were collected, even though in low numbers but they must also be tested for the pathogen. For interest sake were there any other pathogens tested or any co infection observed.

Response: The manuscript has been further edited appropriately. We were unable to test other ticks species and other pathogen because of paucity of funds.

Reviewer #2: 

Comment 1- Nusirat Elelu., et al, reported in the present manuscript entitled: “Prevalence of tick infestation and Molecular characterization of spotted fever Rickettsia massiliae in Rhipicephalus species parasitizing domestic small ruminants in North Central Nigeria” a survey on tick infesting sheep and goats in the Ipata municipal. The authors have collected the tick from the animals at the local slaughterhouse. The authors report that ticks belonging the Rhipicephalus genre were the most collected from the small ruminants. PCR amplification targeting the 350 bp DNA of the Rickettsia 23S-5S intergenic spacer followed by sequencing have revealed the presence of R. massiliae in two Rhipicephalus turanicus ticks. The data collected and presented in the manuscript lack originality. All the results have been already reported in Nigeria regarding the presence of the different tick species identified in the study. In addition, R. massiliae were also reported in Nigeria. Albeit, the authors claim that the manuscript is the first report of the amplification of the DNA of R. massiliae from tick infesting small animals. Even if the primers used in the study targeting the 23S-5S region of Rickettsia were reported able to amplify the DNA of R. massiliae, the shortness of the amplicon (350 bp DNA) is a limitless to compare between related species of Rickettsia. Using other genes like gltA and rpoB were suitable to clarify the genetic position of the species amplified in these study compared to other closet species. Finally, it was suitable that the authors investigate the presence the presence of Rickettsia species in the other tick species rather than Rhipicephalus ticks. Authors can also consider a molecular survey of other related Rickettsiaceae species like Anaplasmataceae to add more valuable data.

Response: Our study appears to be the first in Northern Nigeria and not the entire Nigeria. The amount of funds we had at our disposal was responsible for the scope of the work (Rickettsia massiliae alone with other Rickettsiaceae species not been considered).

Minor revision

Comment 1- Add tick identification methodology to the material and method section.

Response: The methodology for the identification of ticks has been included in the material and method section.

Reviewer #3:

Comment 1 - Why was DNA extracted only in ticks that were identified as Rhipicephalus? Number of ticks is not a scientific reason. Hyalomma ticks can also be infected with Rm.

Response: The low amount of funds received was a limitation to the intended scope of the study to detect for Rickettsia massiliae in all tick species.

Comment 2 - The bootstrap values in the Rickettsia massieliae grouping is really poor. Only 40, which in most cases would not be reported. This may be due to using different lengths of sequences. Were the sequences trimmed to represent the same 350 bp piece amplified in the PCR? Also, how do the authors explain that their potential strain is closer to USA strains than strains in Tunisia.

Response: This has been addressed accordingly.

Comment 3 - The authors should show the alignment and blast results. The authors do not report performing any BLAST analysis, however, this should be the first step before moving into the ClustalW alignment and MEGA analysis.

Response: Report on BLAST analysis has been included in the manuscript. 

Comment 4 - Likewise, the sequences obtained after Sanger sequencing should be deposited into NCBI and the Genbank numbers should be made available to the scientific community.

Response: Sequences obtained has been deposited in the GenBank and the assertion number (OK350078) has been included in the manuscript. 

Comment 5 - The role of sheep as “reservoir” host is a stretch (see minor comments). Unless, several animals are found infected. The significance of two infected ticks in whether an animal may serve as a “reservoir” is unknown. That statement should be deleted.

Response: This statement of sheep as reservoir for Rickettsia massiliae has been deleted.

Minor comments

Comment 6 - Although the issues in grammar and English are not substantial, the reviewer recommends that the authors consult a colleague for editing.

Response: The manuscript has been better edited.

Comment 7 - Lane 75 – rather than regarded, the Rickettsia belong to the SFG or are classified.

Response: This correction has been carried out. 

Comment 8 - Be consistent, sometimes the authors use “μl” and sometimes “ul”.

Response: This has been corrected accordingly. 

Comment 9 - Ticks tend to be seasonal. May the differences reported in this study differ from other studies due to the seasons when they were carried out?

Response: This has been stated in the discussion section.

Comment 10 - Lane 261. Given that only two Rht were collected from sheep and the two were infected with Rm that is a 100% infection rate. By comparing to the rest of the Rhipicephalus sp in the study, the authors are underestimating the infection rates. If fact this study has much higher infection rates that the cited work. Are these sheep from a particular farm? What information is available. Were the two ticks collected from the same animal? And what other ticks were collected from this particular animal? Was it only these two ticks? If so, was the animal infected?

Response: Two of the five Rhipicephalus turanicus collected from sheep were positive for Rickettsia massiliae and the ticks were collected from different sheep sampled at different times. This observation made has been incorporated in the manuscript.

Reviewer #4: 

This manuscript determined the prevalence of tick infestation among small ruminants and molecularly described a clinically significant spotted fever Rickettsia massiliae from Rhipicephalus ticks collected from North-Central, Nigeria, which means regional study of a pathogen and its host.

Comment 1 - Title and keywords title must use different words.

Response: This has been addressed accordingly. 

Comment 2 - Are the sheep where the ticks were found hairless? or not?

Response: No, the sheep were not hairless. 

Comment 3 - Discuss regarding the life cycle of these ticks if it is from one or more hosts?

Response: A brief lifecycle of Rhipicephalus turanicus (the tick positive for R. massiliae) has been added in the manuscript. 

Comment 4 - Concerning the 23S-5S IGS Rickettsia molecular taxonomy it will be important to provide a database on the sequences found.

Response: The sequences found has been deposited in the GenBank and the accession number incorporated in the manuscript. 

Comment 5 - as in the reference in line 19. García-García JC .......,

Response: It has been checked.

Comment 6 - review all references

Response: The references has been reviewed.

---

## [Editor Report · Decision Letter 1]

28 Jan 2022

Prevalence of tick infestation and Molecular characterization of spotted fever Rickettsia massiliae in Rhipicephalus species parasitizing domestic small ruminants in North-Central Nigeria

PONE-D-21-31323R1

Dear Dr. Ola-Fadunsin,

We’re pleased to inform you that your manuscript has been judged scientifically suitable for publication and will be formally accepted for publication once it meets all outstanding technical requirements.

Kind regards,

Brian Stevenson, Ph.D.

Academic Editor

PLOS ONE
---

## [Editor Report · Acceptance letter]

3 Feb 2022

PONE-D-21-31323R1 

Prevalence of tick infestation and molecular characterization of spotted fever *Rickettsia massiliae* in *Rhipicephalus* species parasitizing domestic small ruminants in north-central Nigeria 

Dear Dr. Ola-Fadunsin:

I'm pleased to inform you that your manuscript has been deemed suitable for publication in PLOS ONE. Congratulations! Your manuscript is now with our production department. 

Kind regards, 

on behalf of

Prof. Brian Stevenson 

Academic Editor

PLOS ONE